# Experimental Study on Microchannel with Addition of Microinserts Aiming Heat Transfer Performance Improvement

**Shailesh Ranjan Kumar** [1] **and Satyendra Singh** [2,*]

1   Department of Mechanical Engineering, Motihari College of Engineering, Motihari 845401, India
2   Department of Mechanical Engineering, Bipin Tripathi Kumaon Institute of Technology, Dwarahat 263653, India
*   Correspondence: ssinghiitd@gmail.com; Tel.: +91-945-622-2807

**Abstract:** Microchannel technology rapidly established itself as a practicable solution to the problem of the removal of extremely concentrated heat generation in present-day cooling fields. By implementing a better design structure, altering the working fluids and flow conditions, using various materials for fabrication, etc., it is possible to increase the heat transfer performance of microchannels. Two parameters that affect how well a microchannel transfers heat were only recently coupled, and the complicated coupling of the parameter that affects how well a microchannel sink transfers heat is still not well understood. The newest industrial developments, such as micro-electro-mechanical systems, high performance computing systems, high heat density generating future devices, such as 5G/6G devices, fuel cell power plants, etc., all present thermal challenges that require the use of microchannel technology. In this paper, single-phase flow in microchannels of various sizes, with or without microinserts, is described in terms of its thermal-fluid flow properties, including fluid flow characteristics and heat transfer characteristics considering the compound effects of variations of channel size and addition of microinserts. The trials were carried out using distilled water that had thermo-physical characteristics that varied with temperature. A microchannel with microinserts was developed for managing the high heat generation density equipment. The fluid flow and heat transfer characteristics are explored and analyzed for Reynolds numbers ranges from 125 to 4992, for 1 mm channel size, and from 250 to 9985, for 2 mm channel size. The cooling performance criteria are pressure drop characteristics, heat transfer characteristics, and overall performance, whereas the testing parameters were chosen for the variations in channel size and the addition of microinserts. The influence of inserting microinserts on microchannels is discussed. Results suggest that by inserting microinserts, the performance of the heat transfer of microchannels is significantly improved and, also, fluid flow resistance is increased. The criteria of the thermal performance factor are employed to assess the overall performance of the microchannel. Significant intensification of heat transfer is observed with indication that the addition of microinserts to microchannels and reduction in channel sizes exhibited improved overall performance.

**Keywords:** microchannel; microinserts; heat transfer performance; thermal performance factor

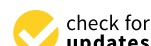



## 1. Introduction

The promising expeditious performance [1] of miniaturized heat-removing devices [2,3] was swiftly established as an efficient and effective cooling system in the modern cooling field. Modern, sophisticated manufacturing techniques allowed for the miniaturisation of innovative, professional equipment that produces an extremely high heat density [4,5] that needs to be removed. Microchannel technology is one such tiny heat-removing technology that is drawn to the challenge of removing tremendous heat density from a restricted zone of equipment with reduced characteristic dimensions [6,7].

The development of microchannel technologies under the leadership of Tuckerman and Pease [8] in 1981 is recognised as a turning point in the field of the contemporary

miniaturised cooling field and drew numerous active researchers to work on improving the thermal-hydrodynamic performance of microchannels. The development of manufacturing processes results in devices with progressively more difficult overall shape and size limits for heat removal. Effective microchannel design improvement becomes essential to overcome these issues. Analyzing the pressure drop, heat transfer, and overall performance is necessary for the efficient design and development of microchannels [9–11].

Numerous studies discussed various factors that affect the performance of microchannels. These performance-influencing factors include the channel's geometry, including its size, shape, and cross-section; the placement of the fluid flow; and the temperature-dependent properties of fluid related to heat transfer [12–16]. Numerous researchers suggested changing the flow of the channels in many different ways to increase the performance of the microchannel [17–22]. Researchers also identified other strategies for enhancing microchannel performance. These include using various working fluids, such as nano-fluids/R134a, altering the material of microchannel's, etc. It was demonstrated that altering the microchannel's structure by adding complexity to it in order to improve performance is an efficient approach to do so. Recently, researchers added complexities to the structure of microchannels in different forms, such as adding pin fins [23–28], cavities [29], ribs [30,31], inserts [32–34], corrugations [35], or a staggered microchannel [36]. However, an insufficient research was done to draw firm conclusions regarding how changing the structure of microchannels may affect their performance. In the past, the majority of works were focused on the investigation of the influences of certain factors, such as either variation in shape [37–40], geometry [41–43], working fluids [44–46], materials [47–49], etc., on the microchannel performance. Very few researchers began to report concerning the study of the compound effect of two or more parameters that influence the performance of the microchannel [50,51].

Microchannel heat sinks deal with thermal challenges associated with the latest industrial developments, such as micro-electro-mechanical systems, including microchips, high-performance computing systems, high-power density batteries, high-power magnets, and high-heat density generating future devices, such as 5G/6G devices, fuel cell power plants, energy, material processing, solar cells and air-conditioning systems, chemical industries, etc. [52–65].

The exploration of the compound effect of change in the structures and cross-section is in nascent age [66]. It requires more study for clear insight into the effect of the compounding of two or more factors. Because of a lack of insights on the influences of multi-factors, it is essential to carry out further investigations on the compound effect of variations in channel size and modifications in the microchannel structures.

It was discovered that the influence on heat transfer gets accompanied with other concurrent compound effects of loss of flow or increase in pump effort as the complexity of the cross-section, as well as the arrangement of the flow, grows. According to past research, it is important to continue investigating the simultaneous combined effects of two or more factors influencing heat transfer performance. The novelty of the present work includes the consideration of the variation in channel size and the addition of microinserts, simultaneously.

By looking at the merit of modifications of structure in the form of pin fins or corrugations or ribs, this paper considered modification of structure in the form of microinserts. In this paper, thermal-hydrodynamic fluid flow features of single-phase fluid flow through microchannels of different sizes, with or without microinserts, is investigated. This paper reports the pressure drop and heat transfer characteristics of microchannels considering the compound effects of variations in channel size and addition of microinserts. This paper defined the cooling performance criteria as pressure drop characteristics, heat transfer characteristics, and overall performance. The testing parameter was chosen as the variations in channel size and the addition of micro inserts.

## 2. Experiment

### 2.1. Experimental Set-Up

For the present paper, single-phase fluid flow through microchannels of various channel sizes with the addition of microinserts is considered. The experimental set-up designed to measure the fluid flow characteristics and thermal performance of the performance of microchannels with microinserts are depicted in Figure 1, which demonstrates the photographic view of the experimental set-up and the test section. The experimental facility's schematic diagram is shown in Figure 2, and it was established to evaluate the microchannel's pressure loss and heat transfer characteristics, as well as its thermal-hydrodynamic properties.

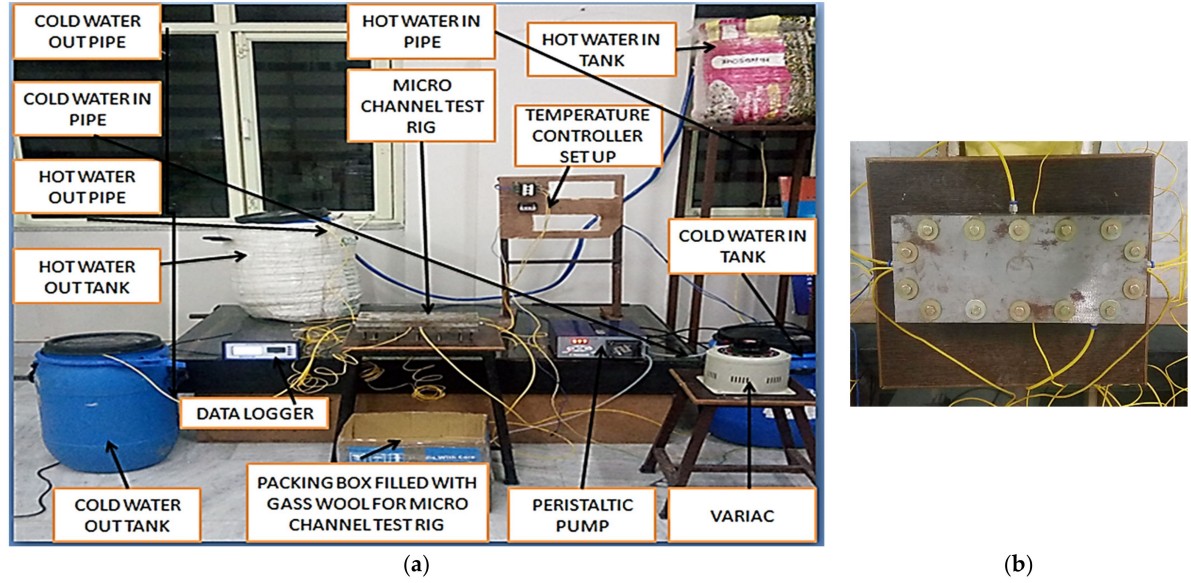

**Figure 1.** Photographic view of (**a**) the experimental set-up and (**b**) test section.

**Figure 2.** The schematic diagram of a flow loop.

The experimental facility can be thought of as consisting of four main subsystems, generally speaking. The heating system might be considered the first subsystem. The second subsystem might be thought of as the management and supply of the working fluid. The test portion could be thought of as the third subsystem. The fourth one may be viewed as the one that gathers information and records output from various sensors as tests are being carried out.

Figure 1 displays an image of the completed experimental test set-up. The experimental set-up comprised of a hot water tank, cold water tank, pump for water circulation, flowmeter, microchannel test section, differential pressure-meter, pipelines, and data logger. Figure 3 depicts the photographic view of a microchannel.

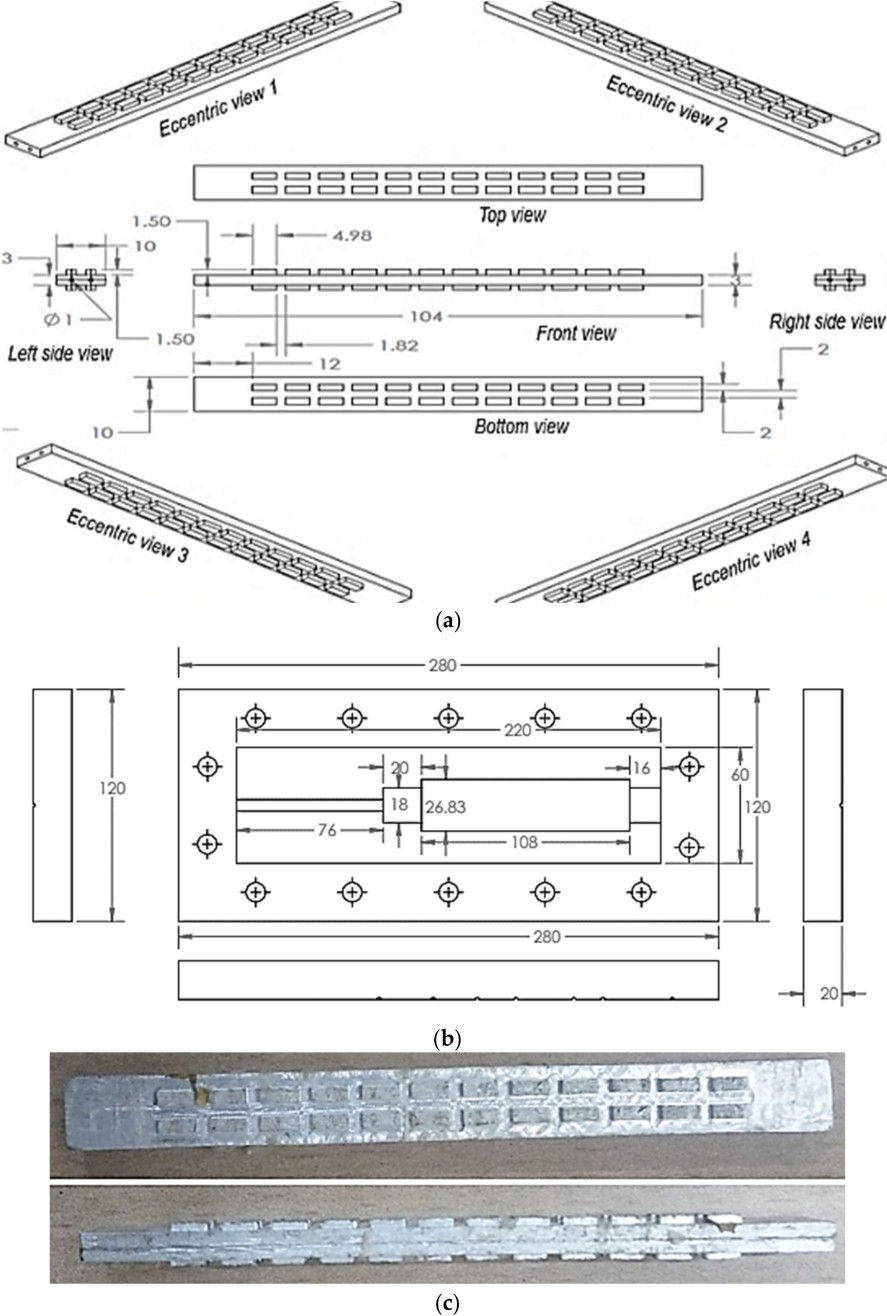

**Figure 3.** (**a**) Geometric parameters, (**b**) centre plate containing microchannel and flow path of hot and cold fluid and (**c**) photographic view of a microchannel with microinserts (unit: mm).



The test modules for microchannels consist of the three components: the centre plate, the heater plate, and the microchannel. Following installation, twelve nuts and bolts are utilised for tight assembly. The top cover plate's bottom surface is pushed up against the top surface of the housing component by the nut bolts. The microchannel under inquiry has a circular cross-section. The microchannel's geometric measurements are 104 mm (L) × 10 mm (W) × 3 mm (H). The microchannel heat sink had complexity added to it in the form of microinserts. The cross-section of the microinserts is rectangular. Microinserts and microchannels both use the same material. Microchannels and microinserts were created together. The rectangular microinserts have the following measurements: 4.98 mm (L) × 2 mm (W) × 1.5 mm (H). The microchannel is made up of aluminium. Experiments were performed and data were obtained at the atmospheric temperature condition of 26 °C.

Seventeen thermocouples in total were utilised to measure different temperatures in the experiments. There were nine thermocouples inserted inside the microchannel test section at the gap of 1 mm, in addition to four thermocouples at the inlet and outlet of hot and cold fluid flow. Four thermocouples were utilised to determine the bulk temperature of the hot water tank, with one touching the top surface and another one touching the bottom layer, and the other two placed throughout the height of the tank with equal spacing. Temperatures were measured with type K thermocouples. The readings of thermocouples were captured during data collection using a data logger that had a direct connection to a personal computer.

## 2.2. Scaling Parameters and Uncertainty

In this section, the parameters [67] defining the thermo-hydraulic characteristics of the microchannel used in the calculation and evaluation criteria are described. The definition of the Reynolds number ($Re$) and hydraulic diameter ($D_h$) at the microchannel inlet are as follows:

$$Re = \frac{\rho U D_h}{\mu} \tag{1}$$

where $\rho$ is the density of the fluid in the unit of kg/m$^3$, $U$ is the fluid velocity in the unit of m/s, $\mu$ is the dynamic viscosity of the fluid in the unit of Pa-s, and $D_h$ is the hydraulic diameter at the microchannel inlet in the unit of mm. The hydraulic diameter is defined in terms of width $W$ and the height $H$, both in the unit of mm, as follows:

$$D_h = \frac{2WH}{W+H}. \tag{2}$$

The pressure drops $\Delta p$ and friction factor $f$ of the microchannel are evaluated as follows:

$$\Delta p = p_{in} - p_{out} \tag{3}$$

$$f = \left(-\frac{D_h}{0.5\rho U^2}\right) * \frac{\Delta P}{L} \tag{4}$$

where $p_{in}$ and $p_{out}$ are pressures measured at the inlet and outlet of the microchannel, respectively, in the unit of N/m$^2$. $\rho$ is fluid density expressed in kg/m$^3$, measured at the arithmetic mean of temperature at the inlet and outlet of the microchannel. Microchannel length is denoted by $L$ in the unit of mm.

The non-dimensional parameter average Nusselt number Nu is evaluated as

$$Nu = \left(\frac{D_h}{k_{f,m}}\right) * \left[\frac{mC_p(T_{b,out} - T_{b,in})}{A_s(T_w - T_b)}\right] \tag{5}$$

where $k_{f,m}$ is the fluid thermal conductivity expressed in W/m-K and measured at the arithmetic mean of temperature at the inlet and outlet of the microchannel. $A_s$ is the contact

surface area, measured in the unit of mm$^2$, at which fluid and microchannel are in contact, and is defined as:

$$A_s = (W + 2H)\, L \tag{6}$$

In this paper, in order to assess the improved performance offered by modified heat transfer surfaces for different-sized microchannels, the treatment of the thermal performance factor (TPF) is adopted. It is defined as a ratio of the heat transfer coefficient of the modified microchannel to that for the plain microchannel, keeping pumping power (pp) the same [68]. Therefore, TPF is evaluated as

$$TPF = \left.\frac{h}{h_s}\right|_{PP} = \left.\frac{Nu}{Nu_s}\right|_{PP} = \frac{\frac{Nu}{Nu_s}}{\left(\frac{f}{f_s}\right)^{\frac{1}{3}}} \tag{7}$$

The uncertainty in primary measurement forms the basis of computation of uncertainties for the various quantities. The maximum uncertainties in the measurements of voltage and temperature are about $\pm 0.5$ V and $\pm 0.2$ °C, respectively. The uncertainty associated with the measurement of pressure drop and friction factor are about 4% and 6%, respectively.

## 3. Results and Discussion

### 3.1. Pressure Drop Characteristics

For viscous fluid flow through the channel, there is fluid resistance offered because of the viscosity of fluid, resulting in gradual decrement of the fluid pressure. Figure 4 shows variation in the pressure drops and friction factor with the Reynolds number. Clearly, values of pressure loss of channel with microinserts were significantly more than that of in channel without microinserts. The pressure loss curve is found to be almost linearly incremental with Reynolds number, except at lower values. The pressure drop with microinserts always shows a greater pressure drop than that of without microinserts. It was found that microchannel with microinserts induces 8–32%, for 1 mm, and 5–103%, for 2 mm, higher pressure drops than without microinserts. Additionally, a microchannel with microinserts has a greater friction factor than without microinserts. This is due to reason that microinserts induce more disruptions in fluid flow at increased flow velocity, causing larger frictional pressure loss due to high resistance to fluid flow. This results in high consumption of pump power for the microchannel with microinserts compared to without microinserts. The pressure drops observed in the test section are because of the flow friction (f) along the liquid flow path of the channel. The pressure drops increase approximately linearly with increasing fluid velocity. Figure 4 shows that at higher Reynolds number pressure drop of a channel with microinserts was significantly larger than that of a channel without microinserts. This is because of the increment in vortices formation due to increasing Reynolds number, leading to higher flow resistance. For a smaller Reynolds number, the values of the pressure drop of channels with and without microinserts were was found to be close to each other. However, at the higher Reynolds number, there was a sharp increment in the value of the pressure drop of the channel with microinserts with respect to the channel without microinserts.

### 3.2. Heat Transfer Characteristics

Variation in different parameters with a mass flow rate for a 1 mm and 2 mm channel size is shown in Figure 5. It presents the variations of the outlet temperature, heat transfer rate, overall heat transfer coefficients, and effectiveness with the mass flow rate in the channel. It depicts the relationship between hot–cold side outlet temperatures (for both with and without microinserts) and cold side mass flow rates. It illustrates the thermal behavior of microchannels with or without microinserts. In the present study, the experiments are performed for two different channel sizes of 1 mm and 2 mm, with conditions of with and without microinserts. The primary factors affecting a heat exchange device's thermal

performance are linked to important factors, including the total heat transfer coefficient, the number of transfer units (NTU), effectiveness, etc.

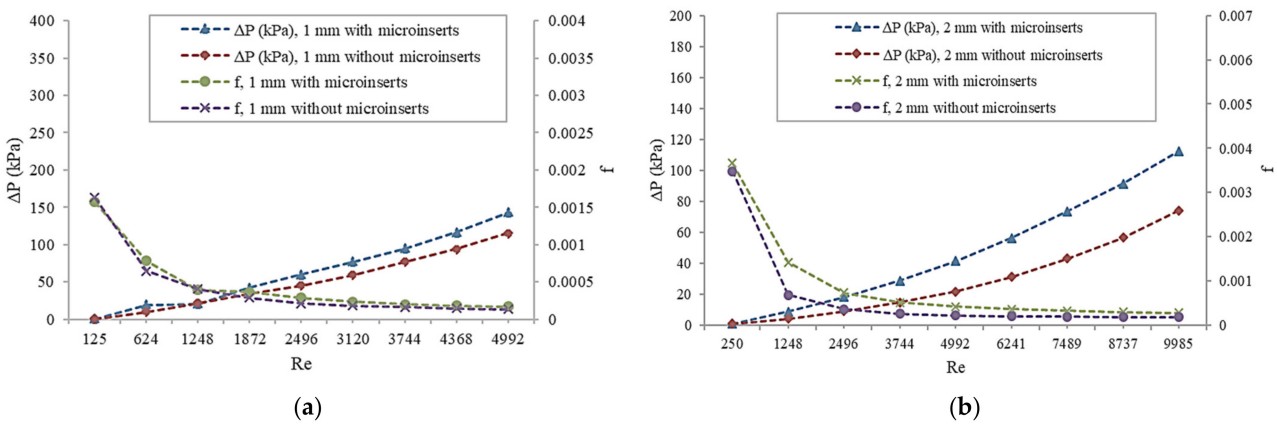

(**a**)                                                                                    (**b**)

**Figure 4.** Variation in pressure drop and friction factor against Reynolds number (**a**) for a 1 mm channel and (**b**) for a 2 mm channel.

(**a**)                                                                                    (**b**)

(**c**)                                                                                    (**d**)

(**e**)                                                                                    (**f**)

**Figure 5.** *Cont.*

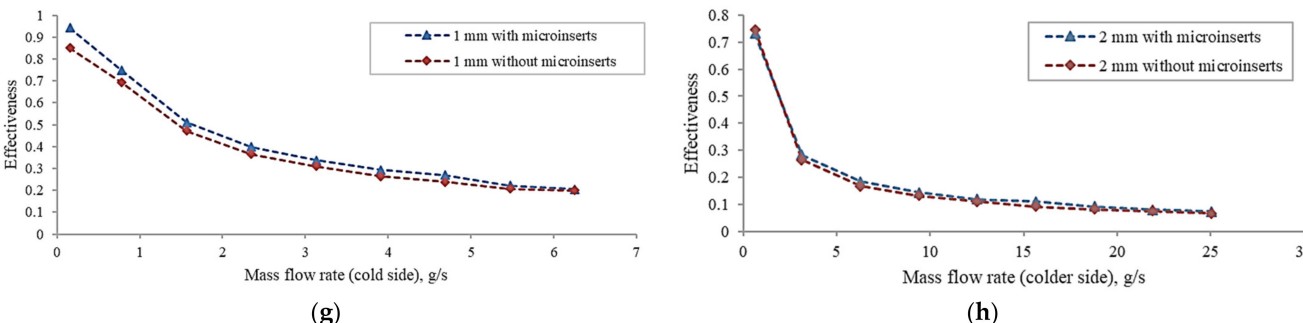

**Figure 5.** Variation in different parameters with mass flow rate for 1 mm and 2 mm channel size.

Using Newton's law of cooling, the overall heat transfer coefficient is calculated in this study and is as follows:

$$Q = UA(\Delta T)_{LMTD} \tag{8}$$

where $Q$, $A$, and $(\Delta T)_{LMTD}$ are heat transfer rate, surface area, and log mean temperature difference, respectively.

The inlet temperature of the cold side and hot side were maintained at 30 °C and 50 °C, respectively. For the hotter side, constant mass flow rate was fixed at 0.782 g/s and 3.127 g/s for 1 mm and 2 mm, respectively. For the colder side, the mass flow rate varied from 0.156 g/s to 6.255 g/s for 1 mm, and from 0.625 g/s to 25.02 g/s for the 2 mm channel size. A comparison of results obtained from experimental data for 1 mm and 2 mm stated conditions is presented in Figure 5. It is found that the outlet temperatures decrease as the mass flow rate of the colder side increase. Since the exchanger considered under study is parallel flow, the temperature of the colder side is always lesser than the outlet temperature of the hot side. It is found that the maximum temperatures for the channel without the microinserts are lesser than that of the channel with microinserts. This indicates that the heat transfer performance of the channel with microinserts was improved.

The overall heat transfer coefficient in the channel with microinserts is significantly higher than that of in the channel without microinserts for both channel sizes. By increasing mass flow rate from 0.16 g/s to 6.25 g/s in the 1 mm channel and from 0.63 g/s to 25.02 g/s for the 2 mm channel, it is found that the overall heat transfer coefficients increased by factor of 1.94–2.41 and 2.03–2.17 for 1 mm and 2 mm, respectively.

Figure 6 depicts plots of the Nusselt number against the Reynolds number for microchannel with and without microinserts. The Nusselt number measures how well a microchannel transfers heat. It measures the proportion of the overall heat transfer to the conductive heat transfer at the fluid flow wall. An increment of the Nusselt number is found because of microinserts at the same value of the Reynolds number. It indicates that the heat transfer rate of the channel with microinserts was better than that of the channel without microinserts. Clearly, the effects of microinserts on heat transfer performance were less pronounced at very low and high ranges of the Reynolds number. The increment in the Reynolds number has an effect of increasing the Nusselt number because of the thin boundary layer formation near the channel walls, resulting in enhanced heat transfer characteristics. The enhanced heat transfer was observed along with an increased pressure drop. However, the high Reynolds number does not allow to have an effective heat transfer between walls of the channel and flowing fluid, as such in a moderate Reynolds number.

The Nusselt number of a microchannel with microinserts is higher than it would be without them, demonstrating how the addition of microinserts improves heat transfer performance. The Nusselt number grows as the channel size decreases. While it becomes flatter as the channel size grows, the slope of the increase in the Nusselt number with the rising Reynolds number is steeper for small channels. This suggests that the performance of heat the transfer can be considerably improved by using narrower channels. Increased convective area and improved flow disturbances are the reason for these improvements in heat transfer characteristics. At two extremes of Reynolds number values, it is discov-

ered that the influence of microinserts on heat transfer enhancement is less pronounced. Microinserts significantly enhance heat transfer properties when Reynolds numbers are in the moderate range for each instance. Effective heat transfer improvement, due to the addition of microinserts, is hindered at very high Reynolds numbers, in contrast to moderate Reynolds numbers. The addition of microinserts to the microchannel improves heat transmission performance.

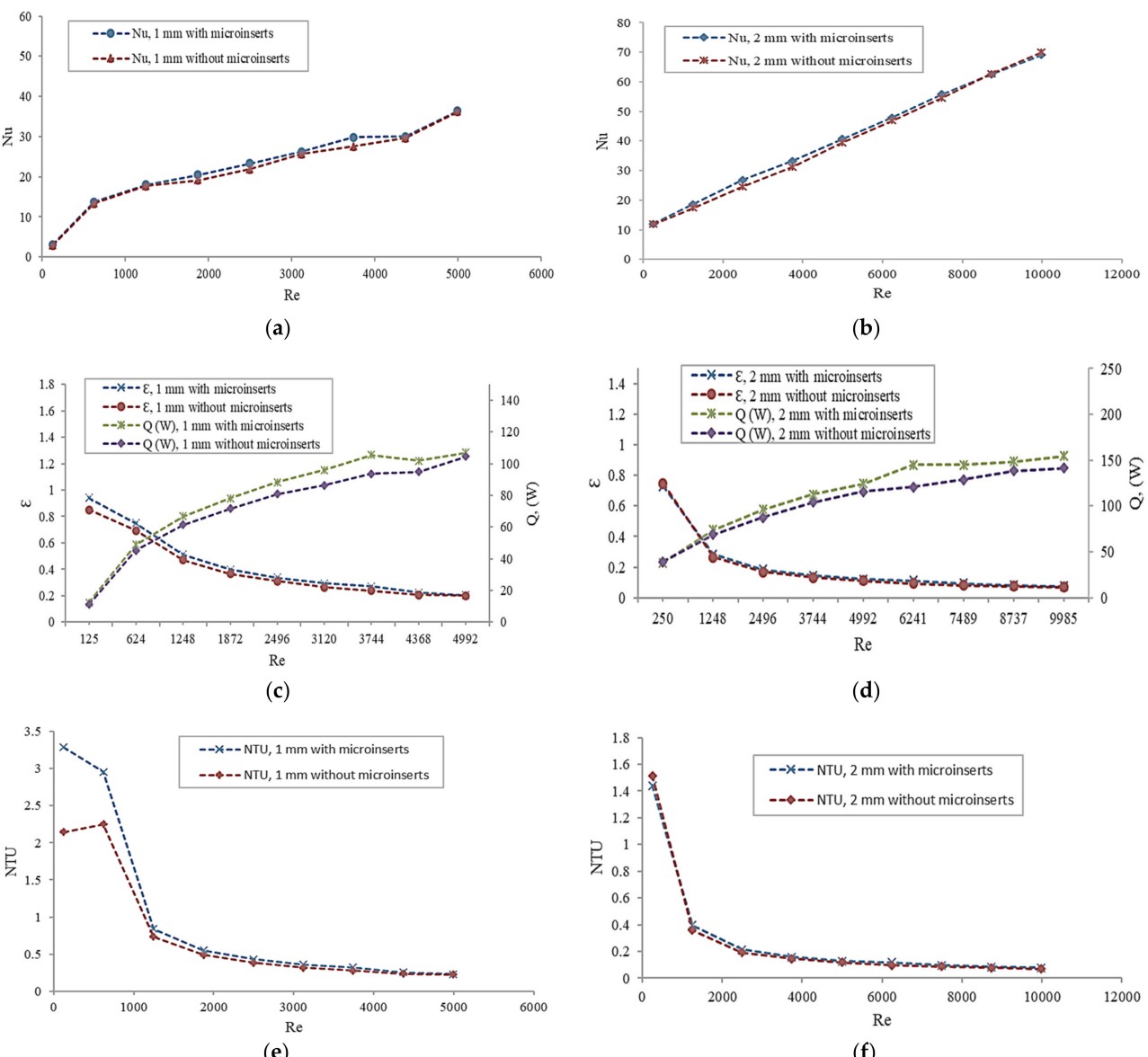

**Figure 6.** Variation in different thermal parameters with the Reynolds number.

It is also found that for smaller channel sizes, as the Reynolds number rises, the Nusselt number rises initially abruptly and then gradually. It implies that the heat transfer between the fluid and the channel wall occurs initially more quickly and thereafter more slowly. This demonstrates that as the Reynolds number rises, the temperature difference between the fluid and the channel wall first decreases quickly before slowing down. By disrupting the thermal barrier layer, the inclusion of microinserts leads to more consistent temperature gradients in fluid flow fields, which is advantageous to enhancing heat transfer. Thus, the insertion of microinserts to microchannels enhances the efficiency of the thermal energy transfer.

### 3.3. Overall Performance Evaluation

The analyzed result shows that microinserts improve the heat transfer performance with the increased resistance to the fluid flow. This necessitates the evaluation of the overall performance of the microchannel. To assess the overall performance, criteria of thermal performance factor (TPF) are used, which is defined as the ratio of heat transfer increment to friction factor increment.

Figure 7 shows the plot of the friction factor ($f/f_s$) and heat transfer performance enhancement factor ($Nu/Nu_s$) against the Reynolds number, where $f_s$ and $Nu_s$ are the friction factor and Nusselt number, respectively, for the plain microchannel. It is clear that as the Reynolds number increases, the values of $f/f_s$ are found to increase. However, rate of increment of $f/f_s$ is low. At an equal Reynolds number, values of $f/f_s$ for a microchannel with microinserts are higher than that without microinserts. The $f/f_s$ of a microchannel with microinserts are found to be in the range of 0.003–0.014 and 0.014–0.044 for channel sizes 1 mm and 2 mm, respectively, for the whole Reynolds number. From Figure 7, it is evident that at the same Reynolds number, $f/f_s$ for the microchannel with the microinsert is larger as compared to plain microchannel. This result suggests that flow resistance for the channel without microinserts gets more influenced by the decrement of dynamic viscosity due to the increase in fluid temperature than that for the channel without microinserts.

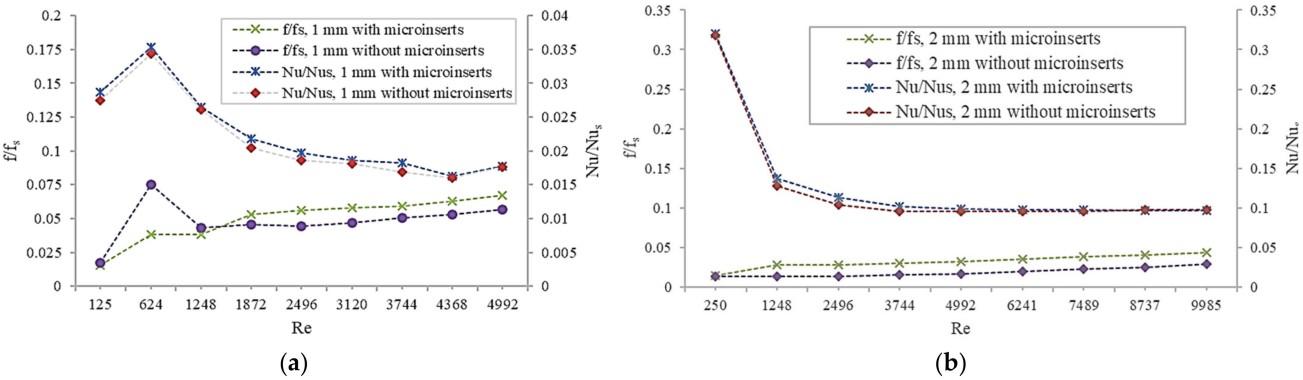

**Figure 7.** Variation in $f/f_s$ and $Nu/Nu_s$ with Reynolds number for (**a**) 1 mm and (**b**) 2 mm.

Figure 7 demonstrates that the values of $Nu/Nu_s$ first increase and then continuously decrease with an increasing Reynolds number for the 1 mm channel. An increasing Reynolds number value of $Nu/Nu_s$ continuously decreases for the 2 mm channel. It is found that the variation trends of $Nu/Nu_s$ with the Reynolds number are different for a 1 mm size. For 1 mm, $Nu/Nu_s$ values increase quickly at first, then decrease sharply, and later decrease slowly with an increase in the Reynolds number. At last, at a higher Reynolds number, it is observed to be an increment in $Nu/Nu_s$. For the 2 mm channel, the decrement is sharp and then decreases slowly, and at last the value nearly approaches a limiting value. This suggests that for the 2 mm channel, for a higher Reynolds number, enhancement of performance is not quite sensitive to increments in the Reynolds number. This means an increment in flow rate will not enhance heat transfer performance further.

Additionally, Figure 7 brings noticeable attention to the fact that for any Reynolds number, $Nu/Nu_s$ values for 2 mm channels with or without microinserts are almost the same and there is little variation in the values of $Nu/Nu_s$ for a 1 mm channel with or without microinserts. This concludes that thermophysical properties that depend on temperature have influence on the heat transfer performance for lower size microchannels. $Nu/Nu_s$ are found to be in the range of 0.081–0.177 and 0.098–0.321 for respective 1 mm and 2 mm channel sizes with microinserts.

For larger channel sizes, values of $Nu/Nu_s$ are seen to be steadily declining with rising Reynolds numbers, with a sharp decline at lower Reynolds numbers and a steady decline at higher Reynolds numbers. Smaller channel sizes respond similarly to big channel sizes throughout a moderate range of Reynolds numbers. $Nu/Nu_s$, however, first

quickly increase at low Reynolds numbers before continuing to slowly decrease. It is discovered that the increase in Nu/Nus is substantially steeper at low Reynolds numbers, as the channel size continues to shrink. Heat transport is influenced by the inclusion of microinserts being simple for all channel sizes. Additionally, as the channel size gets smaller, microinserts become more important because they speed up heat transfer. This tendency is seen in smaller size channels where temperature-dependent thermo-physical features have a greater impact on heat transfer performance. The insertion of inserts has a significant impact on heat transfer enhancement. It can be concluded that the features that are temperature-dependent are crucial for making such observations.

Figure 8 shows a plot of the thermal performance factor (TPF) against the Reynolds number. It is found that the obtained TPF values of the channel with the microinserts are higher than those in the case of the channel without microinserts for the entire Reynolds number. It is found that TPF decreases as the Reynolds number increases, with a sharp decrease found for a lesser Reynolds number and a blunt decrease for a higher Reynolds number for the 1 mm channel. At a lower Reynolds number, for the 2 mm channel, there is a sharp decrement of TPF and then it decreases very slowly as the Reynolds number increases. This concludes that enhancement in performance is more dominant at a lesser Reynolds number, whereas at a higher Reynolds number, the increment of flow resistance is dominant.

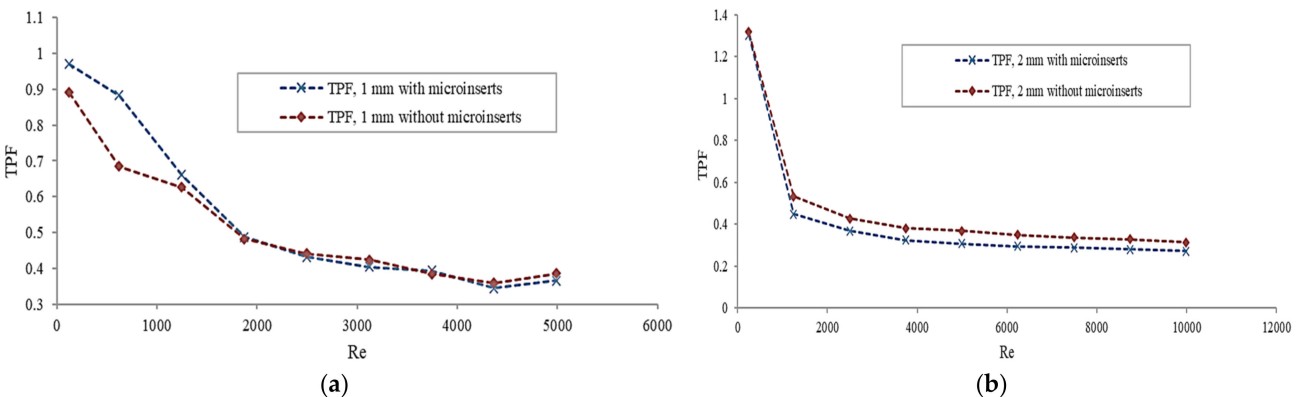

(**a**)  (**b**)

**Figure 8.** Variation in TPF with Reynolds number.

The comparison of the results with the available data (without inserts) is presented in Figure 9. The Reynolds number and the Nusselt number data are taken from published articles of the experimental work conducted [31] and used for comparison. The present findings were discovered to be in agreement and consistency with the reference data. Additionally, the Nusselt number is seen to follow a similar pattern. This examination indicates good reliability of the present proposed model.

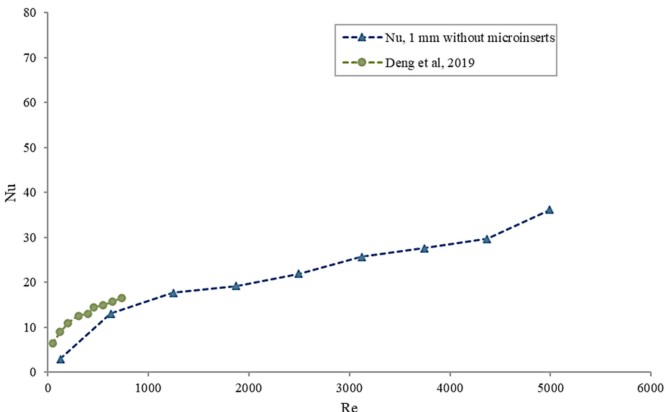

**Figure 9.** Comparison of the results with the available data (without inserts).

## 4. Conclusions

This paper experimentally investigated the thermo-hydraulic performance of the circular channel of two different sizes with or without rectangular microinserts. Obtained results were compared with channels without microinserts. The main findings are summarized as follows:

- The microinserts in the channel resulted in higher fluid outlet temperatures, causing lower base temperature when compared with the channel without microinserts.
- Microinserts performed in enhanced heat transfer, however, also caused a larger pressure drop. Pressure drops of the channel with microinserts were increased by a factor of 1.01–1.32 and 1.05–2.08, corresponding to 1 mm and 2 mm, respectively. The presence of microinserts resulted in increased flow resistance. It was obvious that temperature-dependent thermo-physical properties influenced the flow resistance.
- The heat transfer coefficients, effectiveness, NTU, and Nu of channels with microinserts were found to be increased, as compared to that of the channel without microinserts. The values of Nu were found to be larger by a factor of 1.01–1.08 in the case of the 1 mm and 1–1.07 for 2 mm channel sizes. It is indicated that the thermal performance of channels with microinserts improved. Microinserts effectively enhanced the heat transfer performance for both channel sizes.
- The performance evaluation criteria were employed to assess the overall performance of different channels. The results obtained by this method concluded that the overall performance of the channel with microinserts is better than that for the channel without microinserts for both channel sizes. It was found that microinserts result in the best overall performance at a lower Reynolds number. At a higher Reynolds number, microinserts improve the overall performance only marginally.

In conclusion, this paper investigated the compound effects of variation in channel size and addition of microinserts simultaneously to get an insight of thermo-hydrodynamic behaviors of a circular microchannel. The obtained analyzed results show that the presence of microinserts has a significant influence on the performance of the microchannel. The presence of microinserts enhances the heat transfer performance for both sizes of microchannels, with the simultaneous rise of pressure drop as well. However, overall performance is improved because of the addition of microinserts to the microchannel, as well as with decreasing channel sizes.

**Author Contributions:** Conceptualization, S.R.K.; methodology, S.R.K.; software, S.R.K.; validation, S.R.K.; formal analysis, S.R.K.; investigation, S.R.K.; resources, S.R.K.; data curation, S.R.K.; writing—original draft preparation, S.R.K.; writing—review and editing, S.S.; visualization, S.R.K.; supervision, S.S.; project administration, S.R.K. All authors have read and agreed to the published version of the manuscript.

**Funding:** This research received no external funding.

**Institutional Review Board Statement:** Not applicable.

**Informed Consent Statement:** Not applicable.

**Data Availability Statement:** Not applicable.

**Conflicts of Interest:** The authors declare no conflict of interest.

## Nomenclature

| Symbols | Descriptions | Unit |
|---|---|---|
| $A_s$ | Contact surface area of the fluid and microchannel | $mm^2$ |
| $c_p$ | Specific heat of water | J/kg-K |
| $D_h$ | Hydraulic diameter | mm |
| f | Friction factor | |
| H | Height of the microchannel | mm |

| h | Heat transfer coefficient | W/m$^2$-K |
| $k_f$ | Thermal conductivity of fluid | J/s-m-K |
| $K_s$ | Solid thermal conductivity | J/s-m-K |
| L | Length of the microchannel | mm |
| m | Mass | kg |
| Nu | Nusselt number | |
| p | Pressure | Pa |
| Re | Reynolds Number | |
| T | Temperature | K |
| TPF | Thermal performance factor | |
| U | Fluid velocity | m/s |
| W | Width of the microchannel | mm |
| $\Delta p$ | Pressure difference | |
| $\Delta T$ | Temperature difference | |
| *Greek symbols* | | |
| $\rho$ | Fluid density | kg/m$^3$ |
| $\mu$ | Dynamic viscosity | Pa-s |
| *Subscript* | | |
| *f* | Fluid | |
| *s* | Solid | |

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
