# Peer review of "Experimental Study on Microchannel with Addition of Microinserts Aiming Heat Transfer Performance Improvement"

_water, doi:10.3390/w14203291_

Round 1
Reviewer 1 Report
Comments are provided in the attached file

Author Response
|
S. No. |
Response to reviewers’ comments |
|
1 |
MAIN COMMENT: Uncertainties of the main parameters are not provided. RESPONSE: As suggested, uncertainties of the main parameters have now been added in the revised manuscript. |
|
2 |
MAIN COMMENT: Comparison of the results with the available data (without inserts) of other authors is not provided. RESPONSE: The comparison of the results with the available data of papers of other authors have now been added as per suggestions. We have compared the results from references cited in our paper (Wang et al., 2019 [31]). |
|
3 |
OTHER COMMENT: More detailed description of the test section is necessary. It is necessary to provide the cross section (with dimensions) of the test section with indication of the channels with hot and cold water flows. Information on positions and number of thermocouples also is necessary. RESPONSE: The details have now been added as per suggestions. |
|
4 |
OTHER COMMENT: Formula (2) indicates how Dh was calculated. But in the title, it is indicated that experiments were performed with “Circular Microchannel”. Please clarify. RESPONSE: Dh is an important parameter to consider for microchannel analysis. However, in the case of circular microchannel it is not evaluated as the diameter of circular channel is Dh itself. In the present paper, Dh is of no significance. |
|
5 |
OTHER COMMENT: Please check Fig. 1 and Fig. 3. The photographs are covering some text. RESPONSE: We have adjusted Figures in order to have readable text without covering it. In the word file re-submitted I have assured that it is not covering now.
Thank you very much for your review and comments. They are quite valuable. I have gone through them and improved manuscript as per the suggestions. |
Reviewer 2 Report
Ms. Ref. No.: Water-1936561
Title: Experimental Studies on Circular Microchannel with Addition 2 of Micro-inserts Aiming Heat Transfer Performance Improvement
In this paper, experimental tests were carried out to characterize the thermal-hydraulic performance of a microchannel. A microchannel with micro inserts was developed for managing the high heat generation density equipment. The experimental investigation was performed by employing distilled water as a working fluid. The fluid flow and heat transfer characteristics were explored and analyzed. The influence of inserting micro inserts on microchannel is discussed. Results suggested that by inserting micro inserts the performance of heat 17 transfer of microchannel is significantly improved and also fluid flow resistance is increased. The 18 criteria of thermal performance factors are employed to assess the overall performance of the microchannel. The analyses themselves are sound and I believe the results of their work. The data are well organized by the authors. I, therefore, recommend this paper be published in the Water Journal after the authors address the following comments.
· Review English grammar as there are mistakes throughout the text. This article should be completely rewritten.
· An abstract is not well organized. The abstract must be improved. The authors must explain the application and novelty of the research work add in the abstract section.
· The literature section must be improved with more advanced articles and clearly why your present study is different, better to explain novelty.
· More physical insight into the discussion section is needed.
· The physical explanation of figures 5-7 is limited. Please explain more
· P 9: L 214: This suggests 214 that for 2 mm channel, for higher Reynolds number, enhancement of performance is not 215 quite sensitive to increment in Reynolds number. This means increment in flow rate will 216 not enhance heat transfer performance further. Why?
· P 9:L 226: Figure 8 shows the plot of thermal performance factor (TPF) against Reynolds number. Where is fig 8?!!!!!
· Fig 5 C and D: the authors must be edited the plot based on two lines exactly that they were placed on top of each other. Hot Side?!!
· The authors must add the reference for the equations that are used.
· Nomenclature added to the manuscript.
· The author must improve the introduction with more advanced applications. Also, the author could find new references for the literature review. For example:Chemical Engineering Journal 356 (2019): 492-505, Analytica chimica acta 838 (2014): 64-75. Chemical Engineering Journal 334 (2018): 2603-2615. AIChE Journal 61, no. 6 (2015): 1912-1924. Chemical Engineering Journal 328 (2017): 1075-1086. Separation and Purification Technology 231 (2020): 115875
In conclusion, this paper might be made suitable for publication in this Journal if the as-mentioned comments are clarified. These constitute a major revision of it.
Author Response
|
S. No. |
Response to reviewers’ comments |
|
1 |
COMMENT: In this paper, experimental tests were carried out to characterize the thermal-hydraulic performance of a microchannel. A microchannel with micro inserts was developed for managing the high heat generation density equipment. The experimental investigation was performed by employing distilled water as a working fluid. The fluid flow and heat transfer characteristics were explored and analyzed. The influence of inserting micro inserts on microchannel is discussed. Results suggested that by inserting micro inserts the performance of heat transfer of microchannel is significantly improved and also fluid flow resistance is increased. The criteria of thermal performance factors are employed to assess the overall performance of the microchannel. The analyses themselves are sound and I believe the results of their work. The data are well organized by the authors. I, therefore, recommend this paper be published in the Water Journal after the authors address the following comments. RESPONSE: Thank you very much for your kind words of appreciation! We sincerely appreciate your time and consideration. We have read your comments carefully and tried our best to address them one by one. We hope that the manuscript has been improved accordingly. |
|
2 |
COMMENT: Review English grammar as there are mistakes throughout the text. This article should be completely rewritten. RESPONSE: We agree with the reviewer’s assessment. Accordingly, throughout the manuscript, we have revised/deleted/modified/restructured the words/sentences wherever needed. As suggested, the English grammar mistakes throughout the text have now been rectified. The language of whole paper has been reviewed and modified by rewriting it. |
|
3 |
COMMENT: An abstract is not well organized. The abstract must be improved. The authors must explain the application and novelty of the research work add in the abstract section. RESPONSE: As suggested, the abstract of the paper has been improved by rewriting in the organized form. The application and novelty of the research has now been explained and added in the abstract. |
|
4 |
COMMENT: The literature section must be improved with more advanced articles and clearly why your present study is different, better to explain novelty. RESPONSE: As suggested, the literature section is improved with more advanced articles added. It has also been explained about the novelty of the present study. |
|
5 |
COMMENT: More physical insight into the discussion section is needed. RESPONSE: We think this is an excellent suggestion. We have added the suggested contents to the manuscript. |
|
6 |
COMMENT: The physical explanation of figures 5-7 is limited. Please explain more. RESPONSE: As suggested, results are discussed in more details in order to get more physical insight. The physical explanation of figures 5-7 has now been added in details. |
|
7 |
COMMENT: P 9: L 214: This suggests that for 2 mm channel, for higher Reynolds number, enhancement of performance is not quite sensitive to increment in Reynolds number. This means increment in flow rate will not enhance heat transfer performance further. Why? RESPONSE: By addition of micro-inserts, for higher Reynolds number, enhancement of performance is not quite sensitive to increment in Reynolds number, due to reason of in effective heat transfer between wall and fluid. This means, by the addition of micro-inserts, increment in flow rate will not enhance heat transfer performance further when compared with that of in case of without micro-inserts. |
|
8 |
COMMENT: P9: L 226: Figure 8 shows the plot of thermal performance factor (TPF) against Reynolds number. Where is fig 8?!!!!! RESPONSE: Thank you for pointing this out. The reviewer is correct, and, as suggested, we have added Figure 8 in the manuscript. |
|
9 |
COMMENT: Fig 5 C and D: the authors must be edited the plot based on two lines exactly that they were placed on top of each other. Hot Side?!!. RESPONSE: The heat transfer rate variation of hot side fluid in case of 1 mm with micro-inserts ranges from 3.50 to 4.42 W whereas in 1 mm without micro-inserts ranges from 3.43 to 4.0 W. In case of 2 mm with micro-inserts ranges from 13.47 to 20.01 W whereas in 2 mm without micro-inserts ranges from 15.04 to 19.4 W. Due to marginally changed ranges it appears to be the plot based on two lines exactly and looked like o be placed on top of each other. However, two lines are completely different. |
|
10 |
COMMENT: The authors must add the reference for the equations that are used. RESPONSE: Thank you for pointing this out. We agree that this is an important consideration. We have made changes in manuscript as per suggestions. |
|
11 |
COMMENT: Nomenclature added to the manuscript. RESPONSE: Thank you for the nice reminder. We have added a nomenclature table. |
|
12 |
COMMENT: The author must improve the introduction with more advanced applications. Also, the author could find new references for the literature review. For example: Chemical Engineering Journal 356 (2019): 492-505, Analytica chimica acta 838 (2014): 64-75. Chemical Engineering Journal 334 (2018): 2603-2615. AIChE Journal 61, no. 6 (2015): 1912-1924. Chemical Engineering Journal 328 (2017): 1075-1086. Separation and Purification Technology 231 (2020): 115875. RESPONSE: As suggested by the reviewer, we have modified introduction and referred these suggested papers. |
|
13 |
COMMENT: In conclusion, this paper might be made suitable for publication in this Journal if the as-mentioned comments are clarified. These constitute a major revision of it. RESPONSE: Thank you very much for your comments that helped us to improve this manuscript. |
Round 2
Reviewer 1 Report
No comments
Reviewer 2 Report
Accept